# The Use of Miniature Specimens to Determine Local Properties and Fracture Behavior of LPBF-Processed Inconel 718 in as-Deposited and Post-Treated States

**DOI:** 10.3390/ma15134724

**Published:** 2022-07-05

**Authors:** Jan Dzugan, Mohsen Seifi, Martin Rund, Pavel Podany, Richard Grylls, John J. Lewandowski

**Affiliations:** 1COMTES FHT a.s., Prumyslova 995, 33441 Dobrany, Czech Republic; martin.rund@comtesfht.cz (M.R.); pavel.podany@cometsfht.cz (P.P.); 2Department of Materials Science and Engineering, Case Western Reserve University, Cleveland, OH 4163, USA; mohsen.seifi@case.edu (M.S.); jjl3@case.edu (J.J.L.); 3ASTM International, Washington, DC 20036, USA; 4Beehive 3D, Inc., Deerfield Beach, FL 33442, USA; r.grills@beehiveind.com

**Keywords:** additive manufacturing, local anisotropy, miniature samples, tensile properties

## Abstract

This paper summarizes the assessment of directional anisotropy in local mechanical properties for Laser Powder Bed Fusion (LPBF) IN-718 bulk samples via the use of miniature samples excised from the bulk for both as-deposited and post-treated states. The quasi-static tensile properties at room temperature are investigated at several different locations along the build direction and at different orientations for both considered states. A comparison between the excised miniature tensile specimens and standard-sized sample results have also been conducted and exhibit very good agreement. Significant anisotropy is present in mechanical properties at different build heights for the as-deposited state, while the post-treated material exhibited more homogenous properties, both along the height and for different sampling orientations. However, significant reductions (e.g., >30%) in the strength (Yield, UTS) along with a significant increase in the reduction in area at fracture is found for post-processed materials. Metallography and fractography analyses were conducted in order to begin to determine the source(s) of this anisotropy for the as-deposited state.

## 1. Introduction

While AM technologies are rapidly evolving, the field of potential applications for AM-produced components is also growing. However, implementation of AM for applications in strength/fracture critical components requires reliable performance [1]. This can be facilitated by a better understanding of the resulting materials’ properties by enabling the appropriate design of AM-produced components [2,3,4]. The significantly different process windows for their production in comparison to well-known classical processes (e.g., casting, forging, rolling, etc.) produces features that are different than those present in conventionally processed materials. These features include porosity, non-equilibrium microstructures, surface layers that behave differently from the core, and property anisotropy along and with respect to the build direction, as well as between builds [5]. An initial step in resolving these issues requires systematic evaluation and quantification. Once understood, such local property variations can be incorporated into AM component designs to improve specific part performance. Alternately, anisotropy can be minimized via optimized deposition processes and/or by subsequent component post processing (e.g., heat treatment, thermomechanical treatments, HIP, etc.) [6].

A better understanding of AM-deposited materials requires the ability to assess the location- and orientation-dependence of mechanical properties with the use of minimum material volume. While various small-sized testing methods are available for such assessments, instrumented hardness-based methods and/or Small Punch Test methods are difficult to correlate with mechanical properties that are applicable in design. Mini-Tensile Tests (M-TT) procedures were originally developed to assess the residual service life of power industry components while providing a broadly applicable test method that does not require correlation coefficients for each material investigated, such as that needed for the Small Punch Test method. The M-TT approach has been verified for multiple materials while the currently used test procedure was developed in previous works [7,8,9,10,11] and is being discussed by the standards community [12]. Ongoing issues related to testing of small-sized specimens, such as specimen size effect, are discussed in the literature [13,14,15,16,17,18,19], while the important effects of grain size in relation to sample size for such sub-sized specimens has also been addressed [20,21].

The present work was conducted to extend the expertise in the field of small-sized specimens testing for AM-deposited materials. The focus of the work presented here relates to the demonstration and quantification of local mechanical anisotropy in an idealized Laser Powder Bed Fusion (LPBF)-processed IN-718 rectangular bar. Bars in the as-deposited and post-treated states are reported here. Local mechanical properties in terms of quasi-static room temperature tensile tests along the build height were determined along with sampling various orientations over the build direction, at each height position. Metallography and fractography were also used in order to begin to understand the source(s) of such property variations.

## 2. Materials, Methods and Results

### 2.1. Processing and Specimen Preparation

An SLM 280 machine (SLM Solutions, Germany) was used to produce 10 mm × 20 mm × 120 mm length cuboidal bars using SLM Solutions pedigree gas-atomized IN-718 spherical powders with an average particle size of 15–45 µm. Chemical composition of the powder is shown in Table 1. A standard SLM raster strategy with an angle difference of 67° was used between adjacent layers to deposit the bars in the manner shown in Figure 1. Details of the tension samples excised from the bend bars are also shown in Figure 2. The main SLM processing parameters included 30 µm layer thickness, 0.12 mm hatch spacing, 200 W laser power, and 900 mm/s scanning velocity, along with the raster strategy described above. In order to determine the local properties and any anisotropy along the bar height, miniature samples (i.e., M-TT) were excised from the rectangular bars, in addition to excising a bulk sample, see Figure 1. Two material states were considered-as-deposited and post-treated. Post-treatment consisted of HIP/annealing at 1220 °C at 207 MPa for 2 h with subsequent rapid cooling. Two bars were examined in the post-treated state designated as ***P*** and ***Q***, in order to assess the effects of this post-processing treatment on property homogenization along the builds. Bars ***P*** and ***Q*** were adjacent bars on the same build plate and were prepared to also check for build homogeneity.

Figure 1 shows the sampling locations for the investigated bar in the as-deposited state. The representative positions analyzed include regions just above the start of the build (i.e., marked ‘0’), at the 1/3 and ½ positions along the bar, and at the end of the bar (i.e., marked ‘1’). The primary directions investigated (using the evolving ASTM standard [12]) were ZXY and XYZ at all 4 height locations, see Figure 1. In addition, samples were excised at the 1/3 position in the YXZ and 45° to the X-Y plane. In the case of post-treated bars, more limited and simplified sampling locations were considered at the bottom, middle and at the top of the bar. Two sampling orientations were investigated for each location in the same way as for the as-deposited bar.

Sample designations in relation to the specific orientations, height positions and appropriate batches are summarized in Table 2.

### 2.2. Mini-Tensile Testing (M-TT)

Details of the M-TT specimen geometry based on previous work [7,8] are shown in Figure 2. Following electrical discharge machining, samples were additionally ground on opposite sides of the flat portions to reduce any possible preparation influences. Tensile tests were performed at room temperature under quasi-static loading conditions. Test conditions followed ASTM E8 requirements at 10^−3^ s^−1^ followed by basic parameter evaluations. Specimen thickness was measured with a micrometre while the other dimensions were measured using an optical measuring system. At least three M-TT specimens per location and orientation were tested with the use of a small-sized linear drive-based testing system (Labortech S.R.O., Katerinky, Czech Republic) with 5 kN capacity and valid calibration certificate. Strain in the course of the miniature tests was measured using Mercury Digital Image Correlation (DIC) system (SOBRIETY S.R.O., Kurim, Czech Republic). Prior to each test, strain calibration with certified calibration blocks was performed while sample preparation for DIC measurements consisted of several steps. Specimens were first cleaned using detergents, then a thin uniform layer of dull acrylic spray was used to cover the sample. A rough DIC pattern was achieved by “light” spraying of a black dull spray (graphite spray) that provided a high-contrast stochastic pattern with very small random distributed spots. In a few cases, full field measurement was performed with appropriate post processing to provide more detailed analysis on the strain distribution over the specimen active length. However, the DIC system was used in video extensometer mode and recorded the distance between two points representing the gauge length in order to enable large data processing. Three specimens per condition were tested, except the standard-size specimens where only one test was conducted due to space limitations. Then, 0.2% offset yield stress (YS), ultimate tensile strength (UTS), uniform elongation at maximum force (UE), elongation (EL) and reduction in area (RA) were evaluated. Dimensions post-test for reduction-of-area were determined with the use of a stereo-microscope. The data were evaluated in terms of engineering stress–engineering strain (here designated as extensometer strain).

Comparison of standard sized round specimen with mini-M-TTs is demonstrated in Figure 3. The stress–strain curves exhibit very good agreement for the M-TT and bulk sample geometries for stress levels approaching the ultimate tensile strength. However, the localized plastic deformation and necking with subsequent fracture occurs sooner in the smaller M-TT specimens. Records obtained from the miniature M-TTs tests are provided for all sampling locations and orientations in Figure 4 for the as-deposited state, clearly demonstrating location and orientation related anisotropy over the as-built part. Detailed information on tensile test behavior in the directions investigated are shown separately for all sampling locations for the as-deposited bar in Figure 5, Figure 6, Figure 7 and Figure 8. These figures clearly demonstrate local anisotropy of tensile properties, except for the central part of the bar, where isotropic behavior is evident. However, the results also demonstrate very good homogeneity for each specific location in addition to the excellent repeatability of the M-TT tests. A summary of the tensile engineering stress–strain curves for post-treated bars are provided in Figure 9 and Figure 10. Heat treatment reduced the strength properties and improved material ductility, while almost completely eliminating the anisotropy and location-dependent properties over the component, particularly for the “Q” bar”. A summary of the properties exhibited along and perpendicular to the build direction for the as-deposited samples are provided in Figure 11 and Figure 12, and for post-treated bars in Figure 13 and Figure 14. Comparison of the detailed results for the M-TT and bulk specimens are provided in Table 3. Test results for all material conditions, sampling locations and orientations are summarized in Table 4 and Table 5.

### 2.3. Microstructure Evaluation and Fractography

The samples for microstructural evaluation were cut from tensile specimens’ shoulders with respect to the deposition planes. Samples were then embedded in resin and processed by means of standard metallographic preparation using grinding and subsequent polishing. The microstructure of IN-718 was revealed with electrolytic etching in an aqueous solution of 60% HNO_3_. Imaging was conducted with the use of Zeiss Axio Observer.Z1m optical microscope (Zeiss, Oberkochen, German) and with the use of a JEOL 6380 (JEOL, Tokyo, Japan) scanning electron microscope (SEM) that was used for microstructure observations as well as for fractography. Section cuts in XY, YZ and ZX planes were prepared by means of standard metallographic procedures and images are provided in Figure 15 and Figure 16.

### 2.4. As-Deposited State

The 3D microscopy images in Figure 15 show the microstructure of IN-718 in three locations of the bar (e.g., start, middle, and end of build). The microstructure is rather coarse-grained and exhibits the remnants of melt pools in the YZ and XZ plane, Figure 15. The remnants of the melt pools exceed several layer thicknesses and typically result from the higher energy of the laser power source used in the LPBF process, since more applied energy produces larger and deeper melt pools [4]. A section perpendicular to the build direction (e.g., XY) reveals some elongated grains aligned about 30° with respect to the X axis, although many grains are parallel to the X axis. This is also reflected in a substantial difference between the microstructure in the perpendicular planes (i.e., YZ and XZ) where the boundaries (i.e., on the XZ plane) between the individual layers form lines. The main difference between the specimens occurs in the appearance of the remnants of melt pools evident on the YZ plane. For example, the images of the previous melt pools visible at the start of the build, see (Figure 15a), are much smaller in comparison to those visible in the middle, Figure 15b), and end of the build, Figure 15c), suggesting heat accumulation during the build.

Fracture analyses on the IN 718 M-TT specimens typically revealed evidence of processed-induced porosity with fracture features centered on internal defects as shown in Figure 17a,b. The fracture mechanism in all samples was transgranular ductile failure, Figure 17c.

### 2.5. Post-Treated

Coarse-grained microstructures resulted from the heat treatment of ***P*** and ***Q***, with removal of the columnar structure (i.e., elongated grains) in the YZ and XZ plane. Section cuts in XY, YZ, and ZX planes are captured in 3D microscopy images in Figure 16. The heat treatment promoted precipitation of carbide particles inside and along the grain boundaries (see Figure 16), while the precipitation outlined the shape of the melt pools as shown from detailed metallographic micrographs in the YZ plane, Figure 18. This was particularly evident in the ***P*** heat-treated specimens, Figure 18a,c,e.

The tensile test fracture surfaces of the Z-oriented heat-treated ***P*** and ***Q*** samples in the middle part of the bar are shown in Figure 19 and Figure 20. The M-TT test results did not reveal any significant anisotropy between the Z and X orientations at any positions along the bar. Fractographic analyses revealed transgranular ductile fractures, while Figure 19c or Figure 20c showed very fine equiaxed dimples. The characteristics of the ductile fracture (i.e., void appearance, size, etc.) was neither dependent on the position of the sample along the bar nor on the testing direction of the samples, while the fracture surfaces exhibited no preferential fracture path.

The scatter in the results could be attributed to both the difference between the cooling rates along the bar, samples’ orientation in the powder bed, and also to the microstructural features such as planar defects and entrapped pores. The effects of loading in the X and Z directions could include stress concentrations that provide an easier path for defect growth and coalescence during testing in the Z direction [22].

## 3. Discussion

The present work has built upon the significant previous work developed at COMTES FHT Inc. regarding the use of miniature M-TT samples to explore the location- and orientation-dependence of properties in a variety of AM-processed as well as conventionally-processed materials [5,7,8,9,10,11] to complement related work ongoing at a number of locations [2,13,14,15,16,17,18,19,20,21,22,23]. In the present work on both as-deposited and post-treated IN 718, the M-TT tests conducted successfully illustrated the mechanical property variations at different locations and orientations in the as-deposited materials, Figure 11 and Figure 12 and these appear to be consistent with the microstructural changes present in those regions, captured in Figure 15. The fine-scale microstructure near the start of the build exhibited the highest strengths, consistent with the faster cooling rate experienced due to the lack of powder bed preheat. The coarsening of the microstructure exhibited along the length of the build is consistent with heat accumulation during the AM process, also exhibited in other systems [1,2,3,4,5,6,7,8,9,10,11,12,13,14,15,16,17,18,19,20,21,22], thereby producing reduced strengths in the M-TT samples. Properties in the XYZ orientation were consistently higher than that along the build direction, ZYX. Fracture surfaces of the as-deposited samples showed multiple instances of processed-induced porosity typical in AM-processed materials [1], and even those conducted within the process window recommended by equipment manufacturers. However, the ductility of the as-deposited M-TT samples remained high, with high values for uniform elongation (e.g., >20%) in addition to significant non-uniform strain (i.e., post necking) as revealed by the very high reductions in area at fracture (e.g., typically > 35%) along with the very fine dimpled fracture surfaces. Despite the observations of both location- and orientation-dependent properties in the as-deposited M-TT samples, the bulk ZYX sample produced very similar properties to the ZYX M-TT samples excised from the 1/3 position, roughly taken from the gage length region of the bulk sample. Similar results have been obtained on other materials [5,7,8,9,10,11] that continue to increase confidence in the use of such M-TT samples to document location- and orientation-dependent properties in samples as well as their potential use in parts.

Post-treatment that included HIP homogenized the mechanical properties across all M-TT samples (Figure 13 and Figure 14) and eliminated the anisotropy exhibited by the as-deposited material. The HIP/heat treatment conditions selected completely removed evidence of the as-deposited solidification structure (i.e., melt pools, columnar structure) while also promoting carbide precipitation along remnants of the melt pool boundaries and increases in the grain size, Figure 16 and Figure 17. These microstructural changes significantly reduced the strengths by >30% while further increasing the uniform elongations experienced to >35% while also increasing the non-uniform strains as evidenced by reductions in area that exceeded 55%. Fracture surface examinations revealed that the HIP/heat treatment process reduced the number of process-induced defects present on the fracture surface while retaining the very fine dimple sizes despite the high values of both uniform and non-uniform strain.

## 4. Conclusions

The present work has successfully used miniature M-TT tension samples excised from bulk samples to investigate the location- and orientation-dependence of properties in comparison to bulk LPBF IN 718 in both as-deposited and post-treated conditions. This approach again provides a useful procedure to illustrate the potential differences in local properties from global properties obtained on bulk samples/parts.M-TT samples excised from the gage length locations of the bulk samples exhibited comparable results to standard bulk samples printed at the same time. The resulting offset yield strengths/UTS obtained for M-TT and standard specimens were 182/674 MPa and 182/660 MPa, respectively. These findings are consistent with previous work of this type on other AM-processed materials.The as-deposited material exhibited significant property anisotropy in the different orientations in addition to both location- and orientation-dependent properties. Difference between the tensile properties along the build direction and perpendicular to the build direction reached 15% for some locations, while samples excised along the build direction generally produced lower properties compared to those excised perpendicularly. Microstructure examination of the as-deposited material revealed coarsening of the columnar microstructure along the build direction, suggestive of heat accumulation during the build. This partially contributed to the differences in M-TT properties both along the build and in different orientations, although relatively homogeneous properties were obtained in the middle of the build. Fracture surface examinations revealed multiple examples of process-induced porosity at various locations on the fracture surface, although the remainder of the fracture surface exhibited very fine (e.g., <1 micron) dimples.Metallurgical examinations of the post-treated samples revealed preferential carbide precipitation along grain boundaries and removal of the columnar microstructure.The post-treated samples exhibited isotropic properties as well as significant reductions (e.g., >30%) in the strength (Yield, UTS) along with significant increases in the reduction in area at fracture for all of the M-TT samples in comparison to as-build state. The HIP post-treatment reduced the number of process-induced defects exhibited in the fractured samples, while the fracture surfaces continued to exhibit very fine (<1 micron) dimples.

## Figures and Tables

**Figure 1 materials-15-04724-f001:**
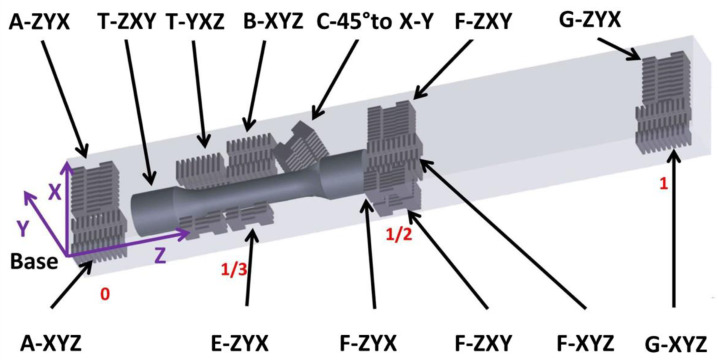
Sample schematics for miniature specimens excised from the as-deposited bar. Various height positions within the bulk bar sample are marked in red (e.g., 0, 1/3, ½, 1, positions). Designation of sample batch, e.g., A-XYZ describes sampling location—‘A’ and samples orientation XYZ in accordance with the ASTM standard [12].

**Figure 2 materials-15-04724-f002:**
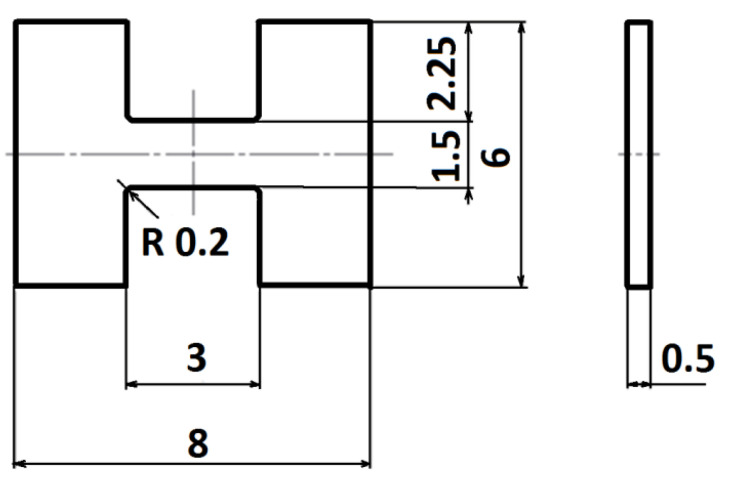
Micro-Tensile Test specimen geometry [7]. Dimensions in mm.

**Figure 3 materials-15-04724-f003:**
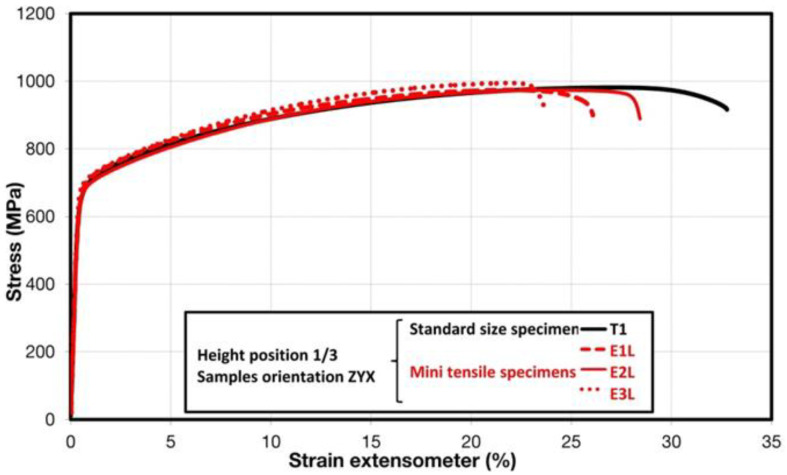
Comparison of standard and miniature M-TT specimen engineering stress–strain plots for AM-produced Inconel 718 in the as-deposited condition.

**Figure 4 materials-15-04724-f004:**
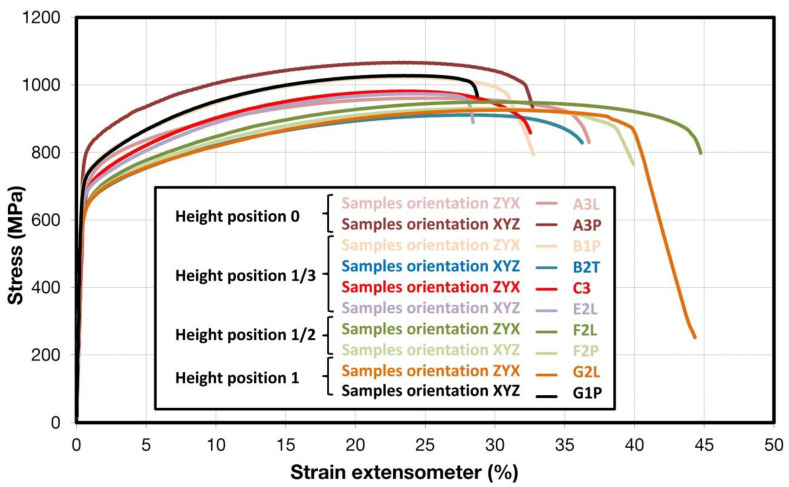
Summary of miniature (M-TT) tensile engineering stress–strain test records for all locations for as-deposited samples.

**Figure 5 materials-15-04724-f005:**
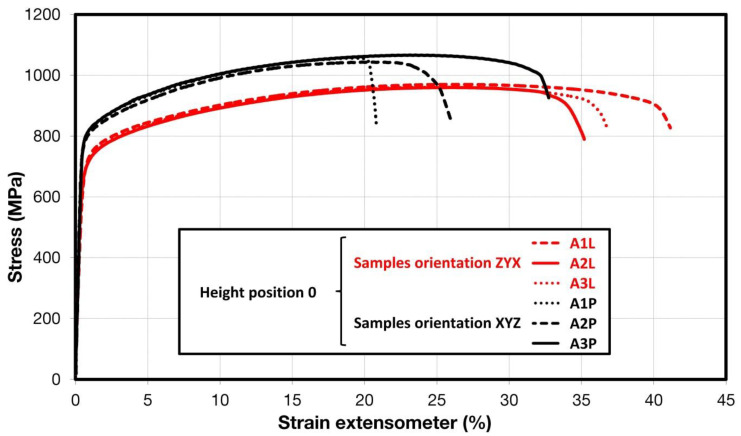
Engineering stress–strain plots illustrating anisotropy at the start of the build in as-deposited samples.

**Figure 6 materials-15-04724-f006:**
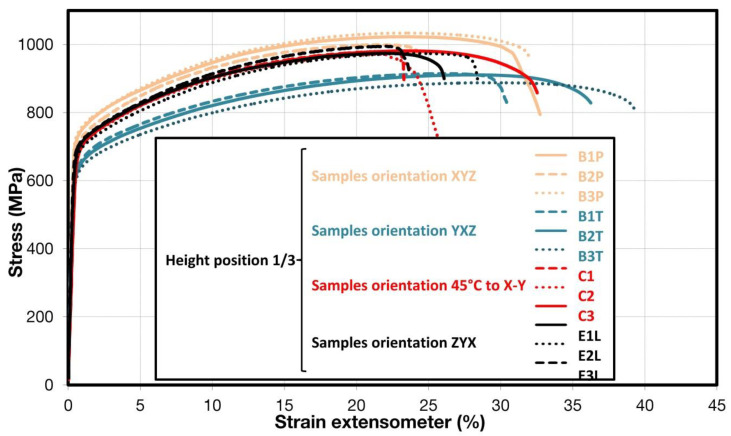
Engineering stress–strain plots illustrating anisotropy at the 1/3 position in 4 different orientations in the build for as-deposited samples.

**Figure 7 materials-15-04724-f007:**
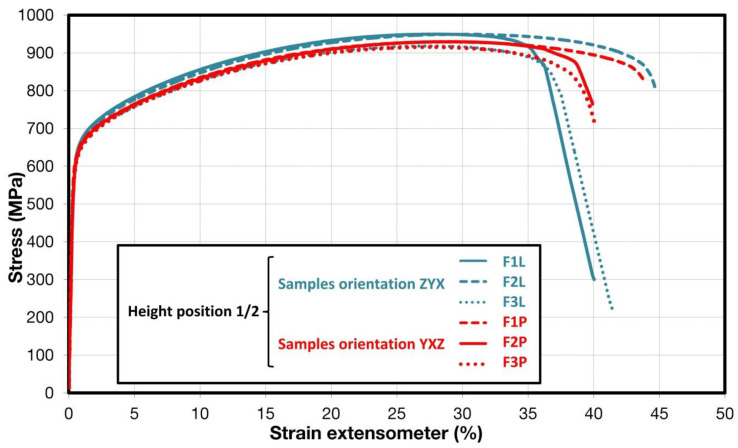
Engineering stress–strain plots showing anisotropy in the middle part of the bar for as-deposited samples.

**Figure 8 materials-15-04724-f008:**
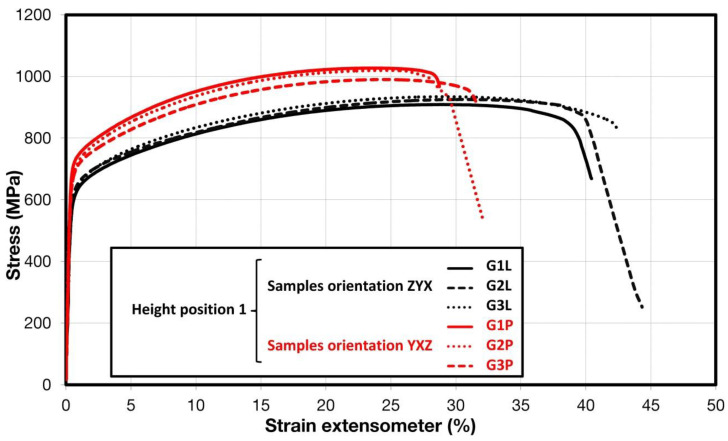
Engineering stress–strain plots showing anisotropy at the end of the build for as-deposited samples.

**Figure 9 materials-15-04724-f009:**
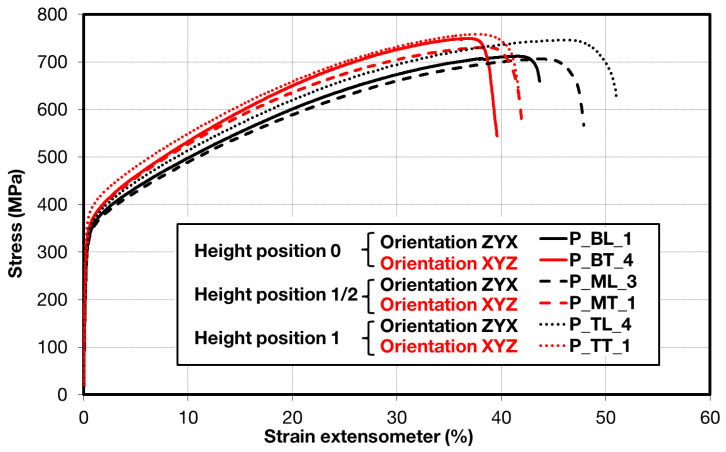
Summary of miniature tensile engineering stress–strain test records for all locations in post-processed ***P*** samples.

**Figure 10 materials-15-04724-f010:**
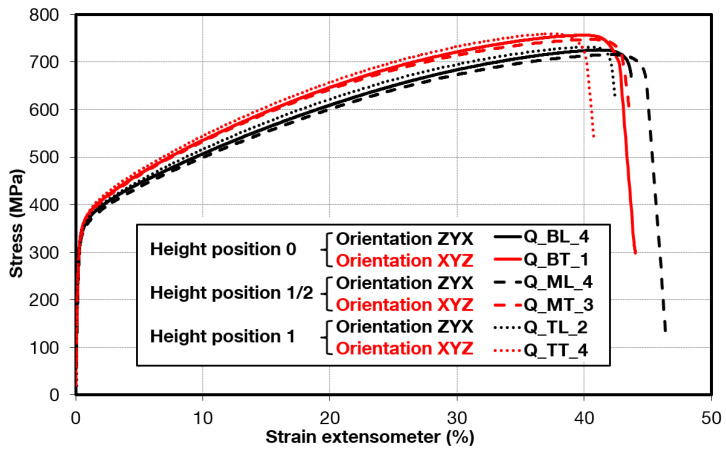
Summary of miniature tensile engineering stress–strain test records for all locations in post-processed ***Q*** samples.

**Figure 11 materials-15-04724-f011:**
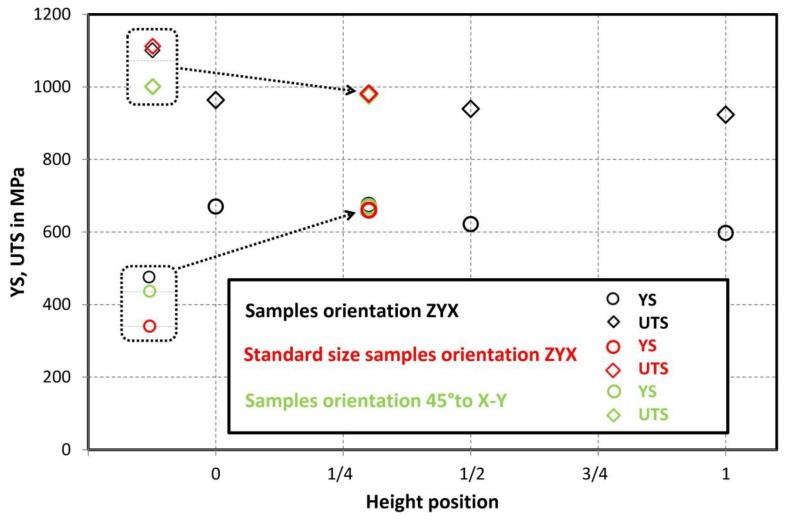
Summary of as-deposited tensile properties obtained along the build direction—samples oriented in the build direction—ZYX in comparison to build orientation—45° to X–Y plane.

**Figure 12 materials-15-04724-f012:**
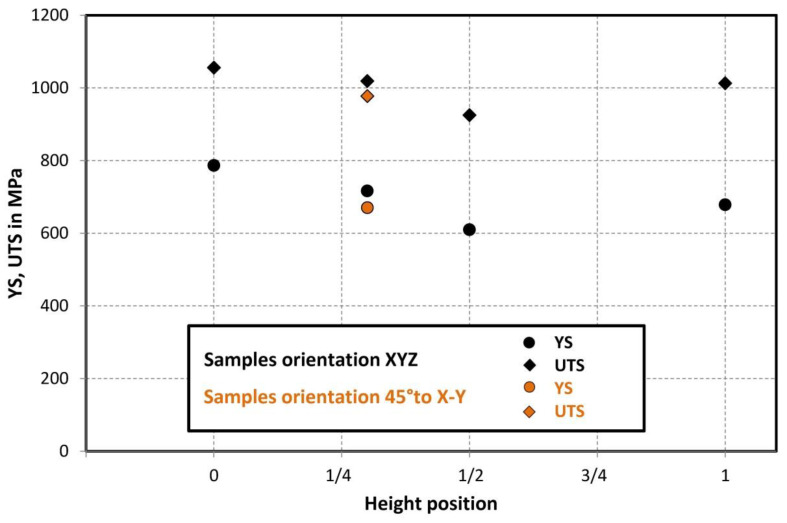
Summary of as-deposited tensile properties obtained along the build direction—samples oriented perpendicularly to the build direction—XYZ.

**Figure 13 materials-15-04724-f013:**
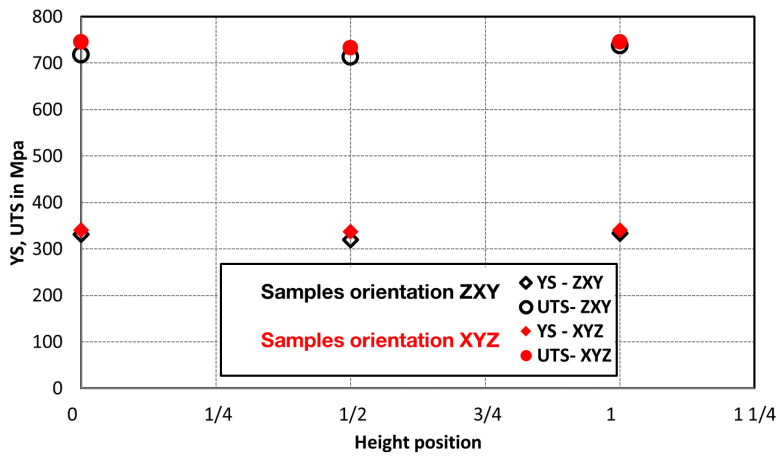
Summary of post-treated tensile properties obtained along the build direction—samples oriented perpendicularly to the build direction for bar ***P***.

**Figure 14 materials-15-04724-f014:**
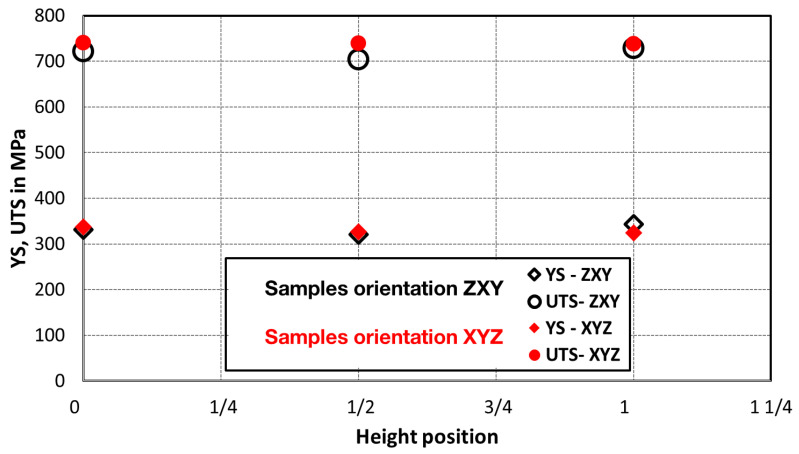
Summary of post-treated tensile properties obtained along the build direction—samples oriented perpendicularly to the build direction for bar ***Q***.

**Figure 15 materials-15-04724-f015:**
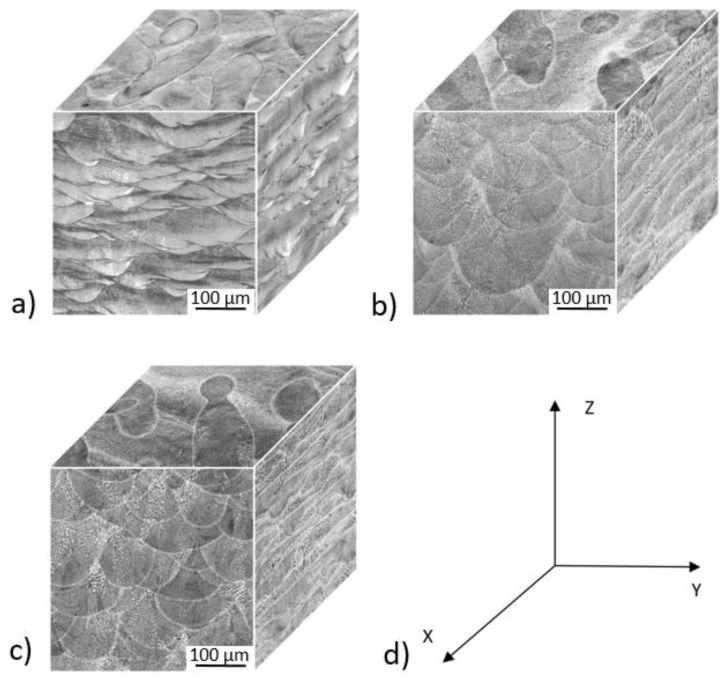
3D metallography showing the front (YZ), side (XZ) and top (XY) microstructure of the Inconel 718 as-deposited bar at the (**a**) bottom (0) (**b**) middle (1/2) (**c**) end top build (1). Figure (**d**) represents the orientation of specimen with respect to the bar where Z is the build direction.

**Figure 16 materials-15-04724-f016:**
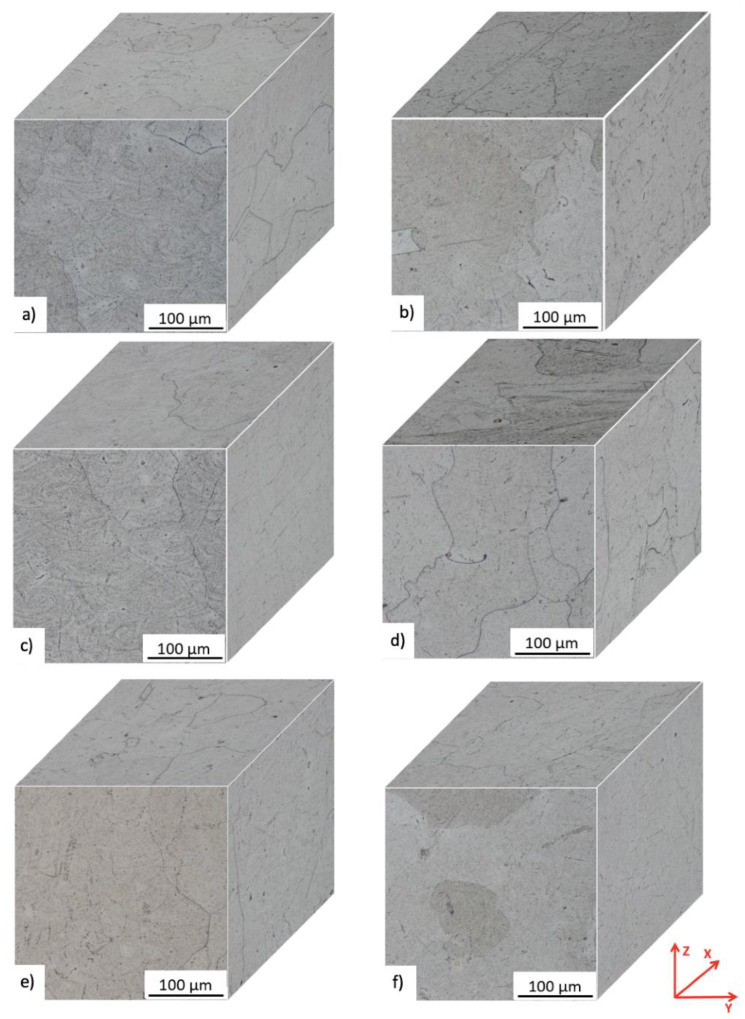
3D microscopy images at bottom, middle and top location for ***P*** and Q post-treated samples. The front (YZ), side (XZ) and top (XY) microstructure of (**a**) P-BT; (**b**) Q-BT; (**c**) P-MT; (**d**) Q-MT; (**e**) P-TT; (**f**) Q-TT. Axis Z represents the build direction.

**Figure 17 materials-15-04724-f017:**
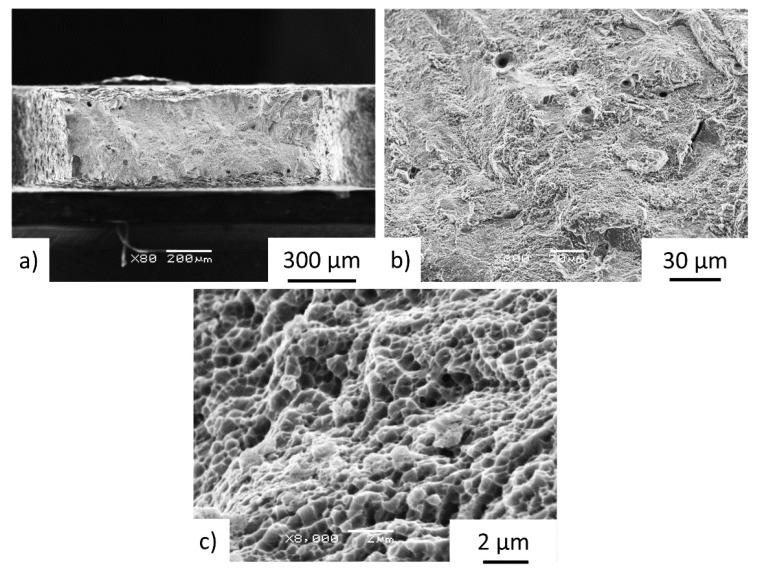
Details of fracture surface of the as-deposited specimen F1–3Z. The fracture surfaces exhibited no preferential fracture path but did show (**a**) presence of processed-induced pores evident at low-magnification (**b**) and at higher-magnification views, and (**c**) high-magnification views showed very fine dimpled features.

**Figure 18 materials-15-04724-f018:**
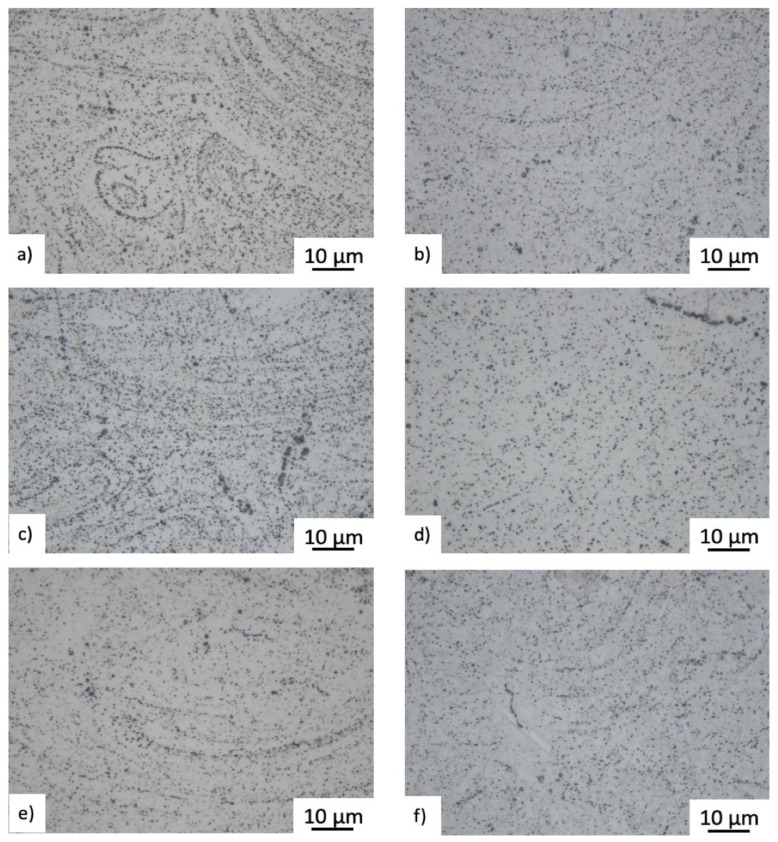
High magnification microstructure views of ***P*** and ***Q*** post-treated samples at bottom, middle and top part of the bar. The front (YZ) plane microstructure of (**a**) P-BT; (**b**) Q-BT; (**c**) P-MT; (**d**) Q-MT; (**e**) P-TT; (**f**) Q-TT.

**Figure 19 materials-15-04724-f019:**
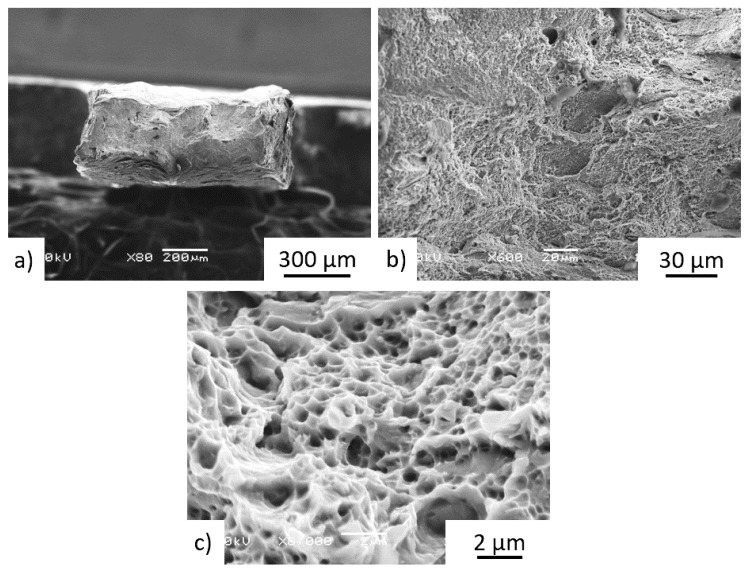
Details of fracture surface of Z-oriented post-treated specimen P-ML3 showed (**a**) ductile fracture with (**b**) transgranular ductile features in addition to more limited process-induced porosity, along with (**c**) fine micro-voids.

**Figure 20 materials-15-04724-f020:**
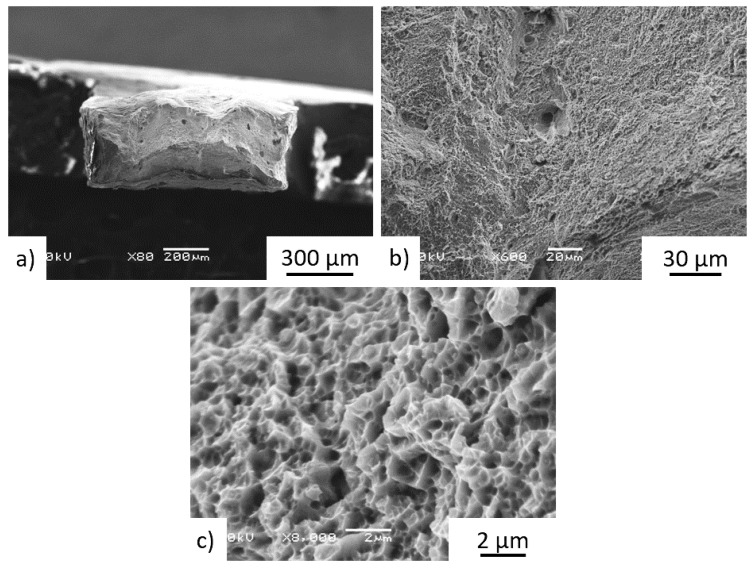
Details of fracture surface of Z-oriented post-treated specimen Q-ML4 showed (**a**) ductile fracture with (**b**) transgranular ductile features in addition to more limited process-induced porosity, along with (**c**) fine micro-voids.

**Table 1 materials-15-04724-t001:** Chemical composition of powder (SLM Solutions specification).

Ni	Cr	Fe	Ta + Nb	Mo	Ti	Al	Cu	C	Si	Mn	B	Co	P	S
50.0–55.0	17.0–21.0	Bal.	4.75–5.5	2.8–3.3	0.65–1.15	0.2–0.8	0.3	0.08	0.35	0.35	0.006	1.00	0.015	0.015

**Table 2 materials-15-04724-t002:** Specimen designations, locations and orientations for as-deposited and post-treated samples. “x” represents the specimen number within the batch. ***P*** and ***Q*** designate two separate adjacent bars that were post-treated.

State	Batch	Height Position	Orientation	Samples Designation
**As deposited**	A_L	0	ZYX	AxL
	A_P	0	XYZ	AxP
	B_P	1/3	XYZ	BxP
	B_T	1/3	YXZ	BxT
	C	1/3	45° to X-Y	Cx
	E_L	1/3	ZYX	ExL
	F_L	1/2	ZYX	F1_xL, F2_xL, F3_xL
	F_P	1/2	XYZ	FxP
	G_L	1	ZYX	GxL
	G_P	1	XYZ	GxP
	T	1/3	ZYX	Tx
**P- post-treated**	P_B_L	0	ZYX	PBxL
	P_B_P	0	XYZ	PBxP
	P_M_L	1/2	ZYX	PMxL
	P_M_P	1/2	XYZ	PMxP
	P_T_L	1	ZYX	PTxL
	P_T_P	1	XYZ	PTxP
**Q- post-treated**	Q_B_L	0	ZYX	QBxL
	Q_B_P	0	XYZ	QBxP
	Q_M_L	1/2	ZYX	QMxL
	Q_M_P	1/2	XYZ	QMxP
	Q_T_L	1	ZYX	QTxL
	Q_T_P	1	XYZ	QTxP

**Table 3 materials-15-04724-t003:** Tensile test results for LPBF-processed IN-718 bar (Figure 1)—comparison of M-TT specimens and standard-sized specimen excised from the as-deposited bar. Elongation was measured in 2 mm gauge for the M-TT samples and from a 25 mm gauge for the bulk sample.

Specimen	Geometry	E	OYS	UTS	UE	EL	RA
		GPa	MPa	MPa	%	%	%
E1_3L_1	M-TT	183.4	670.4	974.4	21.2	26.0	37.1
E1_3L_2	M-TT	177.6	662.4	973.8	23.7	28.5	36.4
E1_3L_3	M-TT	185.1	689.6	994.9	21.2	23.5	28.4
**Average**		**182.0**	**674.1**	**981.0**	**22.1**	**26.0**	**34.0**
**St Dev.**		**3.9**	**14.0**	**12.0**	**1.4**	**2.5**	**4.8**
T1	Standard	182.4	660.1	981.5	26.5	31.6	44.0

**Table 4 materials-15-04724-t004:** Results of M-TT tensile tests conducted on as-deposited IN 718. Highlighted data are provided for comparison of miniature M-TT samples to standard bulk geometry.

Values	Batch	Orientation	Height Position	YS	UTS	UE	A_3mm_	RA
				MPa	MPa	%	%	%
**Average**	**A_L**	**ZYX**	**0**	**669.8**	**963.6**	**25.1**	**37.5**	**41.6**
**ST. Dev.**				**5.4**	**5.5**	**0.8**	**3.1**	**6.5**
**Average**	**A_P**	**XYZ**	**0**	**786.3**	**1055.5**	**20.4**	**26.3**	**34.5**
**ST. Dev.**				**7.1**	**11.5**	**2.3**	**6**	**2.4**
**Average**	**B_T**	**XYZ**	**1/3**	**609.8**	**904.4**	**27.2**	**36.2**	**45.8**
**ST. Dev.**				**7.7**	**14.5**	**1.7**	**3.3**	**2.6**
**Average**	**B_P**	**YXZ**	**1/3**	**716.2**	**1018.7**	**22.2**	**29.7**	**37.7**
**ST. Dev.**				**14.6**	**17.5**	**1.2**	**4.5**	**6.3**
**Average**	**C**	**45°X-Y**	**1/3**	**670.2**	**977.4**	**21.7**	**29.2**	**32.7**
**ST. Dev.**				**5.6**	**5**	**1.2**	**4.9**	**5.7**
**Average**	**E_L**	**ZYX**	**1/3**	**674.1**	**981**	**22.1**	**26**	**34**
**ST. Dev.**				**14**	**12**	**1.4**	**2.5**	**4.8**
**Average**	**F_L**	**ZYX**	**½**	**621.4**	**939.2**	**27.7**	**42**	**63.5**
**ST. Dev.**				**5.9**	**17.9**	**1.3**	**2.3**	**26.8**
**Average**	**F_P**	**XYZ**	**½**	**609.5**	**924.8**	**28**	**41.3**	**48.7**
**ST. Dev.**				**9.6**	**8.1**	**0.7**	**2.3**	**3.3**
**Average**	**G_L**	**ZYX**	**1**	**597.2**	**923.1**	**29**	**42.5**	**47.8**
**ST. Dev.**				**5.9**	**12.8**	**0.8**	**2**	**2.3**
**Average**	**G_P**	**XYZ**	**1**	**677.8**	**1012.4**	**23.9**	**31.3**	**40.1**
**ST. Dev.**				**28.4**	**19.6**	**0.2**	**2.8**	**4.3**
**T1**	**T**	**Z**	**1/3**	**660.1**	**981.5**	**26.5**	**31.6**	**44.0**

**Table 5 materials-15-04724-t005:** Results of M-TT tensile tests conducted on post-treated material for both ***P*** and Q bars.

Values	Batch	Orientation	Height Position	YS	UTS	UE	A_3mm_	RA
				MPa	MPa	%	%	%
**Average**	**P_BL**	**ZXY**	**0**	**330.9**	**718.5**	**41.0**	**43.2**	**56.2**
**St.Dev.**				**5.2**	**8.9**	**4.2**	**4.2**	**3.4**
**Average**	**P_BT**	**XYZ**	**0**	**340.2**	**745.6**	**36.6**	**39.6**	**61.1**
**St.Dev.**				**4.6**	**4.4**	**1.2**	**1.2**	**4.5**
**Average**	**P_ML**	**ZXY**	**1/2**	**319.8**	**713.0**	**43.4**	**47.0**	**61.3**
**St.Dev.**				**1.6**	**8.2**	**1.4**	**2.2**	**1.7**
**Average**	**P_MT**	**XYZ**	**1/2**	**336.8**	**733.3**	**37.9**	**40.8**	**61.6**
**St.Dev.**				**6.6**	**21.1**	**1.7**	**2.4**	**0.8**
**Average**	**P_TL**	**ZXY**	**1**	**333.3**	**737.8**	**44.6**	**49.4**	**57.7**
**St.Dev.**				**4.3**	**11.5**	**1.6**	**2.2**	**3.8**
**Average**	**P_TT**	**XYZ**	**1**	**340.6**	**745.6**	**36.2**	**39.4**	**58.2**
**St.Dev.**				**19.2**	**15.4**	**1,3**	**2.0**	**2.3**
**Average**	**Q_BL**	**ZXY**	**0**	**331.2**	**721.9**	**42.0**	**44.7**	**53.6**
**St.Dev.**				**5.7**	**13.5**	**2.9**	**3.9**	**4.3**
**Average**	**Q_BT**	**XYZ**	**0**	**337.5**	**740.2**	**40.0**	**43.2**	**60.3**
**St.Dev.**				**10.8**	**16.0**	**3.3**	**3.8**	**2.3**
**Average**	**Q_ML**	**ZXY**	**1/2**	**320.7**	**704.1**	**40.3**	**43.5**	**58.2**
**St.Dev.**				**4.0**	**9.0**	**2.8**	**2.8**	**6.0**
**Average**	**Q_MT**	**XYZ**	**1/2**	**327.0**	**739.4**	**40.0**	**43.1**	**54.4**
**St.Dev.**				**4.7**	**5.7**	**0.6**	**1.5**	**2.4**
**Average**	**Q_TL**	**ZXY**	**1**	**343.5**	**728.3**	**39.5**	**42.1**	**57.0**
**St.Dev.**				**12.7**	**11.5**	**2.4**	**2.7**	**2.2**
**Average**	**Q_TT**	**XYZ**	**1**	**324.4**	**737.5**	**37.1**	**39.4**	**57.5**
**St.Dev.**				**14.7**	**23.4**	**2.9**	**3.0**	**6.2**

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
