# Peer review of "The Use of Miniature Specimens to Determine Local Properties and Fracture Behavior of LPBF-Processed Inconel 718 in as-Deposited and Post-Treated States"

_materials, 2022, doi:10.3390/ma15134724_

Round 1

Reviewer 1 Report

In this manuscript, authors investigated a feasible of using miniature M-TT specimens for SLMed IN718 samples’ performance testing. This work is interesting and has good scientific scientific application value. However, I think that this manuscript still needs some improvement. The comments are the following.

Comments:

1. English language and style need to be careful spell check and grammar check.

2. Ensure all figure and table citations in the text match the files provided, such as Figs. 3,11, Table2.

3. In Abstract section, it is advised that major conclusions should be drawn, e.g. UTS, EL, etc.

4. In Introduction section, the background should describe the research progress of miniature samples in 3D printing to show the innovation in this research.

5. In Figure 1, there are a large number of marks for samples, e.g. A-ZYX, T-ZXY, etc., while the naming rules are lacking.

6. Please introduce the scanning strategy in detail. Because the standard raster scanning pattern with an angle difference of 67â—¦ between the two adjacent layers will reduce the trend of anisotropic trends of mechanical properties for SLMed samples.

7. In Line 56, the “laser powder” should be “laser power”.

8. In Line 61, what is the difference between P and Q? Please describe in detail.

9. In Figure 1, please describe the reasons of selecting these height positiThons (e.g. 0, 1/3, ½, 1, positions). Why is there only C-45° sample at position 1/2?

10. In Line 87, the [7][8] is repetitive, please delete it. In Line 88, what is the “EDM”?For the first occurrence, do not use abbreviations. As it is, In Line 242, COMTES?

11. In manuscript, there are a large number of figures that are not described, such as Figs.3-14. The description of experimental results is very important. Please add it.

12. In Discussion, the differences of mechanical properties for different height positions and postprocess need to be highlighted. For Figure 15, it is difficult to discern the size of the microstructure. Please insert high magnification images.

Author Response

All comments are addressed in the attached file.

Reviewer 2 Report

In general, the study was performed at a good level. The presented results of the performed experiments have scientific and practical significance. The research topic is relevant.

In order for the article to be better perceived by the readers, several improvements can be made.

It would be good to make the abstract a little more detailed. Its volume should be about one and a half times larger. 

The introduction (overview part) should be substantially improved, at the moment it is practically absent. Finish the introduction with a general outline of the research carried out in the article and write it more clearly and in more detail.

In the methodology part, it would be good to write the equipment manufacturers and the country of manufacture in parentheses. Please, name the material of the powder used (its chemical composition) so that readers don't have to search on their own.

Lines 84-87. This refers to the introduction section and it would be good to write it in more detail and transfer it to the introduction. Also, it seems unjustified to use such a large number of links 13-21 and 7-11 to confirm the methodology used. 2-3 links are enough. Also, when using links, it would be good to indicate what they contain.

Figures 3-14 show the results of the experiments and it would be good to transfer them from the Methodology section to the Results section. Perhaps it would be necessary to combine or compose such a large number of graphs into a more compact form of representation. The same adice is for tables 2-4.

It would be good to write numerical values of the data obtained in the conclusions and add specifics information (specific values, specific materials).

Author Response

(The authors gave the same response as above.)

Round 2

Reviewer 1 Report

The authors have given detailed replies and modifications. Thus, I suggest that the journal accept this manuscript.

Author Response

Thank you for accepting our revision.

Reviewer 2 Report

The introduction is still small in volume and needs significant revision.

The “Methods” section is not separated from the “Results” section. And on the whole, the manuscript virtually has no structure.

The “Discussion” section is very brief and contains a large number of references and comparisons related to the previous results. There are practically no references to the authors’ own results. There is no discussion for the majority of diagrams.

As a whole, the comments concern the same drawbacks of the manuscript.

Author Response

Reviewer2 comments:

Dear Reviewer, thank you very much for your response, however I am afraid I cannot identify myself with your comments provided in this round of review. During the revision we tried to address all your comments according to our best knowledge and in agreement with our experiences with papers publication in many different high reputation journals, our experiences with reviewing papers and also being editor of the special issue of Materials where this paper should be included. We are fully open and happy to implement useful recommendations to improve our papers, but this second review we do not see as something we can use for the paper improvement.

Response to all specific pointed issues from the review 2:

  • The introduction is still small in volume and needs significant revision.

 It was doubled from previous version and it is one A4 page, this is adequate to the paper considered and there are provide all relevant information, including comprehensive description of work presented in the manuscript. Writing just it is short, is not really helpful, the length is not decisive, but information provided. In the first review there were good advices and we implemented them into the introduction. If something is missing, it should be stated, not general statement: ”not long enough”.

  • The “Methods” section is not separated from the “Results” section. And on the whole, the manuscript virtually has no structure.

 Combination of Methods, Materials and Results is something that is nowadays used. Is there any restriction it should not be sued for “Materials” journal? This combination actually makes quite good sense in the present case as methods  are presented together with results and there is no need to search how results were achieved as all is together easily to follow and understand.

The structure is very clear and straight forward: introduction with background information and work overview description, description of the experimental material and specimens sampling, description of mechanical properties characterization methods together with results, description of microstructure investigations with the obtained results followed by discussion with subsequent conclusions. To me this is quite clear and easy to follow structure.

  • The “Discussion” section is very brief and contains a large number of references and comparisons related to the previous results. There are practically no references to the authors’ own results. There is no discussion for the majority of diagrams.

It is standard practise that achieved results are confronted with previously published results in order to avoid some misleading conclusions based on a wrong data.

The statement:“There are practically no references to the authors’ own results”  - reviewer probably did not realized that many of the used references are results of the submission authors team. So we are referring to our previous results, we are referring to others work and we are also referring to results obtained in the current manuscript, so we think there are all necessary references that are commonly used.

 All summarizing graphs 11-14 are referenced in the discussion and discussed. Graphs with partial results are not discussed, but they are mentioned in the results. In the field of small size specimens testing, there are still many uncertainties and not much transparent data is published, so we are trying to provide these curves to demonstrated clearly how results can look like. First authors is actually chairman of the ASTM group on small size specimens implementation into the ASTM and ISO standard in the AM field and this paper is supporting currently approved standard.

Again used something is "very brief" (in this case discussion), how long should it be?? Is there anything missing?

  • As a whole, the comments concern the same drawbacks of the manuscript.

Another general comment that does not help with the paper improvement.